# Interfaces govern the structure of angstrom-scale confined water solutions

Yongkang Wang [1,2,7], Fujie Tang [3,4,5,7], Xiaoqing Yu[1], Kuo-Yang Chiang [1], Chun-Chieh Yu [1], Tatsuhiko Ohto [6], Yunfei Chen [2], Yuki Nagata [1] ✉ & Mischa Bonn [1] ✉

Nanoconfinement of aqueous electrolytes is ubiquitous in geological, biological, and technological contexts, including sedimentary rocks, water channel proteins, and applications like desalination and water purification membranes. The structure and properties of water in nanoconfinement can differ significantly from bulk water, exhibiting, for instance, modified hydrogen bonds, altered dielectric constant, and distinct phase transitions. Despite the importance of nanoconfined water, experimentally elucidating the nanoconfinement effects on water, such as its orientation and hydrogen bond (H-bond) network, has remained challenging. Here, we study two-dimensionally nanoconfined aqueous electrolyte solutions with tunable confinement from nanoscale to angstrom-scale sandwiched between a graphene sheet and calcium fluoride ($CaF_2$) achieved by capillary condensation. We employ heterodyne-detection sum-frequency generation (HD-SFG) spectroscopy, a surface-specific vibrational spectroscopy capable of directly and selectively probing water orientation and H-bond environment at interfaces and under confinement. The vibrational spectra of the nanoconfined water can be described quantitatively by the sum of the individual interfacial water signals from the $CaF_2$/water and water/graphene interfaces until the confinement reduces to angstrom-scale (<~8 Å). Machine-learning-accelerated ab initio molecular dynamics simulations confirm our experimental observation. These results manifest that interfacial, rather than nanoconfinement effects, dominate the water structure until angstrom-level confinement for the two-dimensionally nanoconfined aqueous electrolytes.

Water under nanoconfinement is the subject of increasing focus[1–6], because nanoconfinement has the potential to modify water properties from bulk water[7–9], and water in nanoconfinement provides a unique platform for chemical reactions[10–12]. For example, water in nanoconfinement exhibits anomalously low dielectric constant[8] and undergoes structural phase transition[4,5,13]. Water in nanoconfinement may exhibit either weakened or strengthened hydrogen bonding[14,15] nevertheless, its hydrogen-bond dynamics are significantly slower than those of bulk water[16–22]. Water and ions under nanoconfinement show extraordinary transport properties[23,24]. Water molecules are more reactive in nanoconfinement than in bulk, spawning the realm of chemistry in confinement[10–12,25]. These anomalous water behaviors

[1]Max Planck Institute for Polymer Research, Mainz, Germany. [2]School of Mechanical Engineering, Southeast University, Nanjing, China. [3]Pen-Tung Sah Institute of Micro-Nano Science and Technology, Xiamen University, Xiamen, China. [4]Laboratory of AI for Electrochemistry (AI4EC), IKKEM, Xiamen, China. [5]Institute of Artificial Intelligence, Xiamen University, Xiamen, China. [6]Graduate School of Engineering, Nagoya University, Nagoya, Japan. [7]These authors contributed equally: Yongkang Wang, Fujie Tang. ✉e-mail: nagata@mpip-mainz.mpg.de; bonn@mpip-mainz.mpg.de

in nanoconfinement are usually ascribed to the nanoconfinement effects. In many experimental studies of the properties of nanoconfined water, deviations from bulk theory expectations already appear at modest confinement (~2–10 nm)[8,22,24,26–36]. For instance, in the case of water in two-dimensional confinement, experimentally observable nanoconfinement effects on water's properties, such as its molecular arrangement[22,26,27,35], dielectric behavior[8,28,31], and transport behavior[24,29,30,32–34,36], appear already at confinement larger than 2 nm. Water's properties highly depend on its molecular arrangement, such as its orientation and H-bond network[37–39]. These experimental observations imply long-range confinement effects on water orientation and H-bond network extending over multiple nanometers[8,22,24,26–36].

On the other hand, extensive theoretical and computational studies examining water orientation and H-bond network have revealed[40–46] the confinement effects typically extend over smaller length scales than multiple nanometers. Specifically, these simulations have proposed distinct confinement- and interface-dependent regimes for nanoconfined water, often defined by critical thicknesses of 2–4 water layers. Below these critical thicknesses, nanoconfinement is expected to induce reorientation of water[4,5,13] and weakened/ strengthened hydrogen bonding strength among water[14,15]. Distinct from the confinement effect, the interface-dependent regimes state that the properties of water under nanoconfinement are only altered by water in contact with two independent interfaces of the two-dimensional confinement[34,40]. Indeed, the structure of interfacial water also differs from bulk water, with a truncated hydrogen bond network and ordered structure due to the interaction with the surface[47,48]. As such, a fundamental and urgent question awaiting experimental examination is at what level of confinement the nanoconfinement effects on water begin to manifest, causing the structure (orientation and H-bond network) of water under nanoconfinement to deviate from the cooperative interfacial effects.

To address the question and differentiate between the interfacial effects and nanoconfinement effects on water, molecular-level insights into nanoconfined water and interfacial water, such as their orientation and H-bond structure, are necessary. To this end, heterodyne-detection sum-frequency generation (HD-SFG) spectroscopy is uniquely suited, owing to three advantages: molecular specificity (in particular, sensitivity to hydrogen bonding strength and molecular orientation); interface and confinement specificity; and additivity of the molecular water response. In particular, the third property allows us to distinguish the additivity of two interfacial effects from true confinement effects. In an SFG experiment, infrared and visible laser fields are mixed to generate the sum frequency of those two fields. The signal is enhanced when the infrared (IR) frequency resonates with the molecular vibration, providing specificity to molecular structure[49]. Furthermore, HD-SFG spectroscopy provides complex-valued $\chi^{(2)}$ spectra and the sign of the imaginary part of the $\chi^{(2)}$ spectrum (Im$\chi^{(2)}$) reflects the absolute orientation of water molecules (up-/down-orientation)[50]. SFG signals are non-zero only when Centro-symmetry is broken, such as at water interface[47,51,52] or nanoconfinement-induced alignment of water[41]. Signals from the bulk water are naturally excluded due to the SFG selection rule[53]. As such, it can selectively distinguish water experiencing the nanoconfinement effects and the interfacial effects. Furthermore, an HD-SFG signal ($\chi^{(2)}$) is additive and thus allows us to quantitatively disentangle the contributions from the two interfaces and the nanoconfinement in the nanoconfined system. With these advantages, one can directly compare the structure of the nanoconfined water with the structure of the interfacial water at an edge of the bulk water, opening a path to distinguish the difference between the interfacial effects and the nanoconfinement effects on water. Nevertheless, applying HD-SFG spectroscopy to the realm of nanoconfined liquids has been challenging owing to 1) the length mismatch between nanofluidic devices and typical laser spots: for extreme (sub-nanometer) confinement, devices are dimensioned below a few hundred nanometers[8,23,54] while the SFG probes a spot of ~100 μm; and 2) the challenge of the IR beam to reach the nanoconfined region.

In this work, we overcome these two challenges by fabricating a centimeter-sized two-dimensional nanoconfined water system with tunable confinement from nanoscale to angstrom-scale sandwiched between a flat calcium fluoride (CaF$_2$) substrate and a graphene sheet. The tunable two-dimensional nanoconfinement is achieved by capillary condensation[55,56] between the hydrophilic CaF$_2$ substrate and the graphene sheet. Furthermore, both the CaF$_2$ substrate and the graphene sheet allow the IR beam to reach the nanoconfined surface. By applying HD-SFG spectroscopy to the two-dimensional nanoconfined water system with tunable confinement and comparing the nanoconfined water response to the response of water at the CaF$_2$/bulk water interface and bulk water/graphene interface, we identify the transition, at angstrom-scale confinement, from water dominated by interfacial effects to that dominated by nanoconfinement effects.

## Results

### Preparation of nanoconfined water via capillary condensation

We prepared the nanoconfined water sample sandwiched by a graphene sheet and a hydrophilic CaF$_2$ substrate by trapping water molecules with Li$^+$ and Cl$^-$ ions under controlled relative humidity (RH) condition (Methods and Supplementary Methods)[2,6,55]. Using this method, the nanoconfined region is composed of the aqueous LiCl solution with a LiCl concentration in the ~few molar ranges (Supplementary Discussion S1). We ensured that the CaF$_2$ substrate is atomically smooth, with a root mean square (RMS) surface roughness close to that of graphite. The flatness of the CaF$_2$ substrate and the homogeneity of the nanoconfined water sample are discussed in Supplementary Discussions S2 and S3. The high quality of the nanoconfined water sample's graphene sheet was verified using Raman microscopy. The Raman data shows the absence of the D-band (~1350 cm$^{-1}$), indicating that the graphene sheet has no obvious defect[57]. The 2D-band and G-band intensity ratio of ~3 confirms that the graphene sheet comprises a monolayer (see Supplementary Discussion S4 for more details).

Subsequently, we measured the thickness of the nanoconfined water using atomic force microscopy (AFM) at a controlled RH ~ 25%. To ensure that the probed sample region was the same for AFM and HD-SFG spectroscopy, a gold structure was used to mark a sample region of ~250 × 200 μm$^2$ at the edge of the graphene sample (Fig. 1a, b, see Supplementary Discussion S3 for more details). An AFM image probing the edge of the graphene sheet is presented in Fig. 1c, while a typical height profile indicated by the white dashed line in the AFM image is shown in Fig. 1d. The edge of the graphene sheet is folded over, producing a sharp height peak. The height difference between the region where the graphene sheet covers and the region where the graphene is absent is 11.5 ± 2.8 Å. By using the estimated thickness of the exclusive volume of the graphene sheet of 3.3 Å from the ab initio molecular dynamics (AIMD) simulation (see Fig. S9 in Supplementary Methods), we conclude that the thickness of the nanoconfined water in our samples is 8.2 ± 2.8 Å. A schematic of the composition of the nanoconfined water sample is shown in Fig. 1e. Furthermore, the AIMD simulation indicates the ~8 Å nanoconfined water is composed of three water layers. The nanoconfined three-layer (3 L) water with its thickness of ~8 Å at an RH of ~25% can be accounted for by the capillary condensation under hydrophilic confinement (see Supplementary Discussion S5 for more details), consistent with previous studies[55,58].

### HD-SFG spectroscopy of nanoconfined water

We carried out HD-SFG measurement within the marked region on the nanoconfined water sample at the ssp polarization combination with the three letters indicating the polarizations of the SFG, visible, and

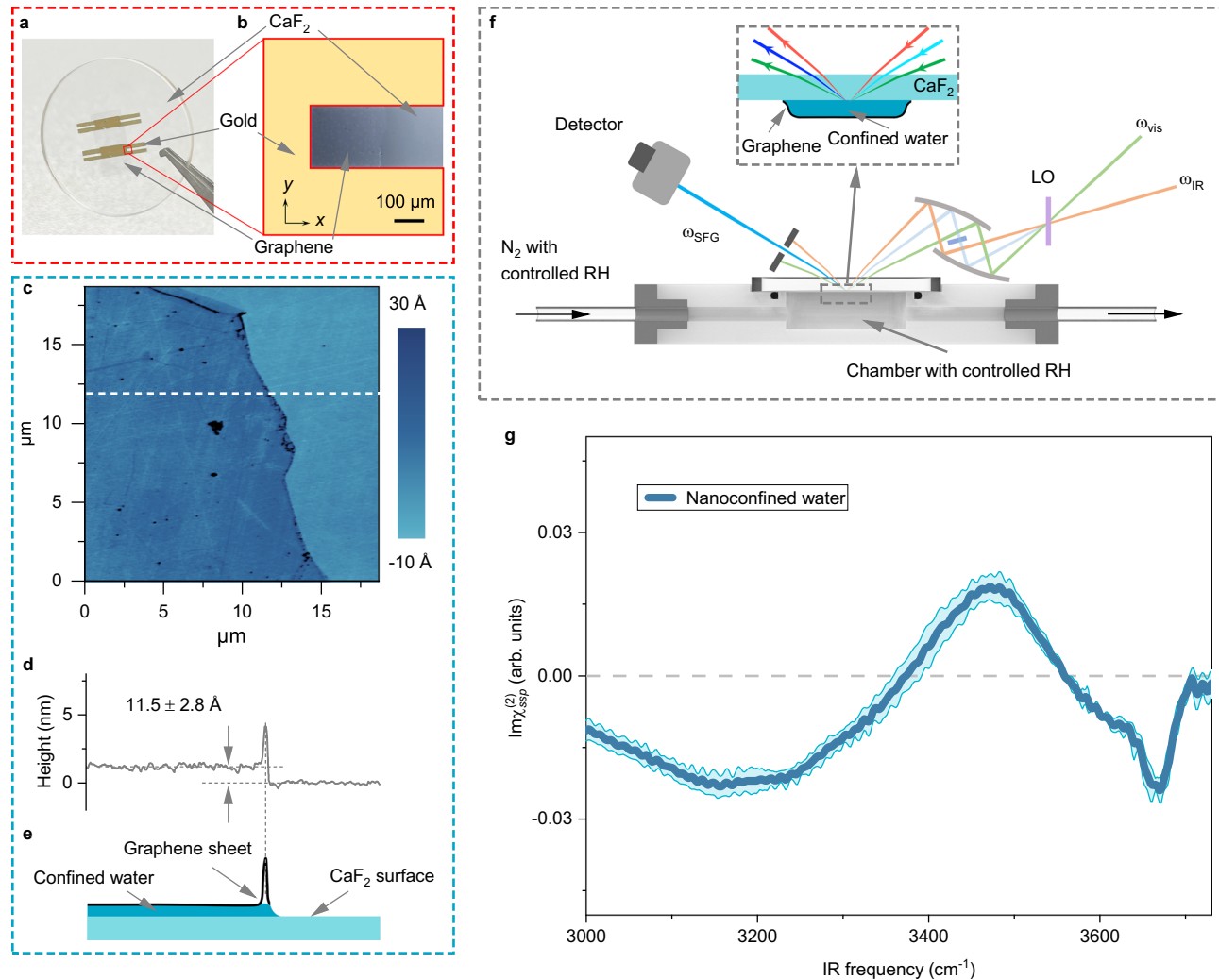

**Fig. 1 | HD-SFG spectroscopy of nanoconfined water. a** An optical image of the nanoconfined water sample surrounded with gold film on a $CaF_2$ substrate. The grey shade area indicates the region covered by graphene. **b** A close-up optical image of the nanoconfined water sample. **c** AFM image of the edge of the graphene sheet of the nanoconfined sample. **d** The height profile along the white dashed line in (**c**), crossing the graphene wrinkles. The error bar represents the standard deviation (S.D.) calculated using height values across the AFM scan area. **e** A schematic of the composition of the nanoconfined sample. **f** Illustration of HD-SFG experiment. **g** The $Im\chi_{ssp}^{(2)}$ spectrum of the nanoconfined water sandwiched between the graphene sheet and the $CaF_2$ substrate. The error bar represents the S.D. calculated from independent measurements on six different samples. The grey dashed line serves as a zero line. SFG sum-frequency generation light, vis visible light, IR infrared light, $\omega$ angular frequency of light, LO local oscillator, RH relative humidity, arb. units arbitrary units. Source data are provided as a Source Data file.

infrared light fields, respectively (see Fig. 1f). Our discussion primarily focuses on the *ssp* polarization combination signal ($\chi_{ssp}^{(2)}$) for straightforward spectral interpretation (see also Supplementary Discussion S6 for measurements with the *ppp* polarization combination). The $Im\chi_{ssp}^{(2)}$ spectrum for the nanoconfined 3 L water system is displayed in Fig. 1g. This spectrum exhibits negative ~3670 cm$^{-1}$, positive ~3460 cm$^{-1}$, and negative ~3200 cm$^{-1}$ peaks. A positive (negative) peak in the O-H stretch $Im\chi_{ssp}^{(2)}$ spectrum corresponds to the up- (down-)oriented O-H group to the $CaF_2$ substrate, with down pointing towards the graphene sheet[47,59–61]. Furthermore, low (high) O-H stretch peak frequencies indicate a strong (weak) H-bond strength of the O-H group[62]. The mixture of the positive and negative peaks spanning over the 3000–3700 cm$^{-1}$ frequency region indicates that these O-H groups with different H-bond strengths are oriented differently.

The interpretation of the SFG spectrum of the nanoconfined water is complicated because two interfaces are probed simultaneously[41]. To uncover the origin of the SFG features for the nanoconfined water system, we compared the SFG spectrum of the nanoconfined 3 L water with the spectra measured at the bulk LiCl solution/suspended graphene interface and the $CaF_2$/bulk LiCl solution interface (see

Supplementary Methods). For brevity, we refer to the LiCl solution as "water" afterward. The quantitative comparison of the water SFG signals from these different samples requires conversion of the measured $Im\chi_{ssp}^{(2)}$ into $Im\chi_{yyz}^{(2)}$ spectra by correcting for Fresnel factors and the local oscillator (LO) reflectivity using the three-layer dielectric model[63,64] (see Supplementary Discussion S7 and S8). The validity of this analysis will be justified below. The inferred $Im\chi_{yyz}^{(2)}$ spectra for the nanoconfined 3 L water system, the water/graphene interface, and the $CaF_2$/water interface ($Im\chi_{yyz,\,confined}^{(2)}$, $Im\chi_{yyz,\,W/G}^{(2)}$, and $Im\chi_{yyz,\,CaF_2/W}^{(2)}$) are displayed in Fig. 2a–c, respectively.

The $Im\chi_{yyz,\,confined}^{(2)}$ and $Im\chi_{yyz,\,W/G}^{(2)}$ spectra both contain the negative 3670 cm$^{-1}$ peak with almost the same amplitude. This peak arises from a dangling O-H group pointing down to the graphene sheet[47,59], weakly interacting with the $\pi$-orbital of the graphene sheet without forming H-bonds with other water molecules[47,59,65]. In contrast, the negative hydrogen-bonded (H-bonded) O-H peak at ~3200 cm$^{-1}$ in the $Im\chi_{yyz,\,confined}^{(2)}$ spectrum is missing in the $Im\chi_{yyz,\,G/W}^{(2)}$ spectrum. Furthermore, the peak amplitude of the positive H-bonded O-H peak at ~3460 cm$^{-1}$ in the $Im\chi_{yyz,\,confined}^{(2)}$ spectrum is much weaker

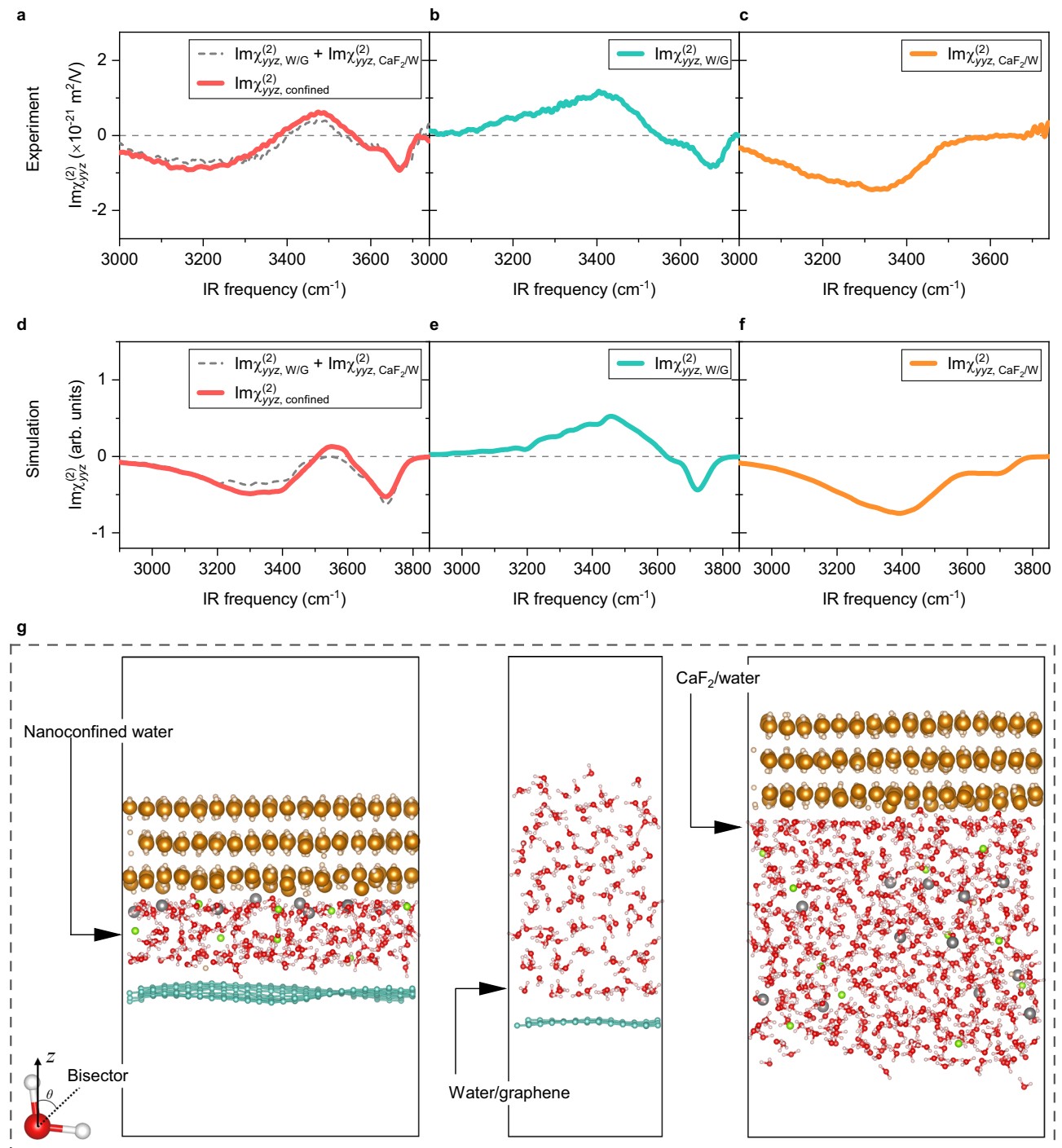

**Fig. 2 | Nanoconfined three-layer water organization is governed by interfacial effects. a** Experimental $Im\chi^{(2)}_{yyz}$ spectra of nanoconfined three-layer (3 L) water (red solid line). The sum of the water/graphene and CaF$_2$/water signals obtained from HD-SFG experiments is also shown for comparison (grey dashed line). **b**, **c** Experimental $Im\chi^{(2)}_{yyz}$ spectra at the (**b**) bulk water/graphene interface and (**c**) CaF$_2$/bulk water interface. We used a 2 M LiCl aqueous solution for the HD-SFG experiments shown in (**b**) and (**c**). **d**–**f** Theoretical $Im\chi^{(2)}_{yyz}$ spectra of (**d**) the nanoconfined three-layer water (red solid line); (**e**) the water/graphene interface; and (**f**) the CaF$_2$/water interface. The sum of the water/graphene (**e**) and CaF$_2$/water (**f**) signals is shown in (**d**) for comparison (grey dashed line). **g** Snapshots of the nanoconfined three-layer water (MLFF-MD), water/graphene interface (AIMD), and CaF$_2$/water interface (MLFF-MD) obtained from the MD simulations. The yellow, light yellow, red, light pink, green, grey, and cyan spheres indicate the Ca, F, O, H, Cl, Li, and C atoms, respectively. The inset in the bottom-left corner illustrates the molecular orientation of a water molecule. The $z$-direction is taken as the surface normal across the interfaces. Dashed lines in (**a**–**f**) serve as zero lines. Source data are provided as a Source Data file.

than that in the $Im\chi^{(2)}_{yyz,\,G/W}$ spectrum. Such discrepancies can be attributed to the impact of the CaF$_2$ surface on the SFG spectrum of the nanoconfined 3 L water. In fact, the $Im\chi^{(2)}_{yyz,\,CaF_2/W}$ spectrum shows the negative H-bonded O-H peak at ~3350 cm$^{-1}$, explaining the appearance of the negative 3200 cm$^{-1}$ peak in the $Im\chi^{(2)}_{yyz,\,confined}$ spectrum. This negative 3350 cm$^{-1}$ peak in the $Im\chi^{(2)}_{yyz,\,CaF_2/W}$ spectrum arises from the interfacial water pointing away from the positively charged CaF$_2$ surface[61].

## Interfacial effects on the structure of nanoconfined water

Remarkably, the SFG signal for the nanoconfined 3 L water can be quantitatively accounted for by combining the water/graphene interface and CaF$_2$/water interface signals, i.e.:

$$\chi^{(2)}_{yyz,\,confined} = \chi^{(2)}_{yyz,\,W/G} + \chi^{(2)}_{yyz,\,CaF_2/W}. \tag{1}$$

The comparison between the Im$\chi^{(2)}_{yyz,\,confined}$ spectrum and the summed spectrum of the two Im$\chi^{(2)}_{yyz,\,W/G}$ and Im$\chi^{(2)}_{yyz,\,CaF_2/W}$ spectra is shown in Fig. 2a. The two spectra overlap within the experimental uncertainty, manifesting that Eq. (1) holds (see also the comparison of the real part of the two spectra in Supplementary Discussion S9). Such consistency suggests that the confinement itself induces no effect on the water organization (orientations and H-bond strength) for the nanoconfined 3 L water, and the organization of the nanoconfined 3 L water is governed by the superposition of the two independent interfaces.

Note that the $\chi^{(2)}_{yyz,\,W/G}$ spectrum is insensitive to ion concentration, whereas the $\chi^{(2)}_{yyz,\,CaF_2/W}$ spectrum at the charged CaF$_2$/water interface is sensitive due to the so-called $\chi^{(3)}$ contribution[66–69]. The $\chi^{(3)}$ contribution arises from long-range electrostatic interactions that polarize water over a characteristic length scale defined by the Debye length[66–69]. Consequently, the ion concentration of the nanoconfined LiCl solution may influence the validity of Eq. (1). In our nanoconfined water system, the charged CaF$_2$ surface establishes a minimum ion concentration (>1 M) required to maintain overall electroneutrality[70]. In fact, our HD-SFG spectral analysis reveals that the ~8 Å nanoconfined LiCl solution has an ion concentration in the range of ~2–3 M (see Supplementary Discussion S1). These high ion concentrations effectively screen the charge on the CaF$_2$ surface, ensuring that the observed changes in the SFG spectra are not due to the truncation of long-range electrostatic effects on water. Instead, despite the charged nature of the CaF$_2$ surface, it is primarily the first layer of water in direct contact with the CaF$_2$ surface that contributes to the overall SFG signal (see Fig. S13 for more details). Importantly, a careful examination of the effect of high-concentration LiCl ions on the organization of the surface water at the charged CaF$_2$ surface reveals a minor dependence on ion concentration at high concentrations (≥2 M, see Supplementary Discussion S1). Thus, within the plausible LiCl concentration range (2–6 M), we conclude that Eq. (1) remains valid, as further confirmed by explicit machine-learning MD simulations discussed in Supplementary Discussion S1. We also examined ion-specific effects by comparing LiCl and KCl and found that the SFG spectra of nanoconfined water are insensitive to ion type. Notably, Eq. (1) also applies to the capillary devices fabricated with graphene and SiO$_2$ (see also Supplementary Discussion S1).

To convert the recorded Im$\chi^{(2)}_{ssp}$ response from the three different samples into experimental Im$\chi^{(2)}_{yyz}$ spectra requires Fresnel factor and the LO reflectivity corrections[63,64]. To do so, we employed the three-layer dielectric model, but the choice of model affects the inferred Im$\chi^{(2)}_{yyz}$ spectra[71,72]. Thus, it is essential to independently validate the Im$\chi^{(2)}_{yyz}$ spectra. To this end, we carried out the SFG spectra simulation for the nanoconfined 3 L water (LiCl solution) sample and CaF$_2$/bulk LiCl solution sample with machine learning force field MD (MLFF-MD) simulations, as well as the bulk water/graphene sample with AIMD simulations (see Methods and Supplementary Methods). The simulation allows us to directly access the Im$\chi^{(2)}_{yyz}$ spectra so the validity of Eq. (1) can be examined critically[64,71].

The simulated Im$\chi^{(2)}_{yyz}$ spectra of the nanoconfined 3 L water system, the water/graphene interface, and the CaF$_2$/water interface are displayed in Fig. 2d–f, respectively. Corresponding snapshots of the MLFF-MD/AIMD simulations are shown in Fig. 2g. The lineshapes of the simulated Im$\chi^{(2)}_{yyz}$ spectra for all these three systems closely resemble the experimental data. Furthermore, the relative peak amplitudes of

the SFG spectra for these three systems also agree with the experimental data. Notably, the amplitude of the positive peak at ~3450 cm$^{-1}$ at the water/graphene interface is close to that of the negative peak at ~3400 cm$^{-1}$ at the CaF$_2$/water interface. Nevertheless, due to the slightly different peak frequencies of the positive peak and the negative peak, the sum of the two SFG spectra creates a negative-positive feature in the 3000–3600 cm$^{-1}$ region (Fig. 2d). Eventually, the summed SFG data agrees well with the SFG data for the nanoconfined 3 L water system, meaning that Eq. (1) also holds for the simulated SFG spectra. This demonstrates that even with its thickness of 3 L water, the nanoconfinement itself induces no unique water organization. The organization of the nanoconfined 3 L water is governed by the interfacial water, manifesting as a simple superposition of the two independent interface systems. Similar interfacial structure of water for the two bulk water samples and the nanoconfined 3 L water sample is also apparent from the joint probability plot of the two O-H bonds and water dipole orientation analysis[73], presented in the Supplementary Information (Supplementary Discussion S10). Furthermore, the Fresnel factor corrections and the LO reflectivity corrections we employed in the experiment are sufficient to provide accurate amplitude calibration for the measured SFG data (see Supplementary Discussion S7 and S8 for more discussions). We also confirm that our phase measurement method does not affect our conclusions (see Supplementary Methods).

It is important to note that, in our implementation, SFG is primarily sensitive to the first water layer at each surface of the CaF$_2$-graphene capillary and largely insensitive to the central water layer in the nanoconfined 3 L water sample. However, since the structure of the water layers near the surfaces remains unaffected, there is no evidence suggesting that the SFG-invisible central water is structurally modified —it can reasonably be assumed to behave as bulk water. This assumption is consistent with previous studies from our group and others, which have established that the interfacial region is confined to a short range, typically 1–2 layers of water[69,74–76]. To further support this, we directly examined the structure of the central water layer in the nanoconfined 3 L water sample using AIMD simulations. Specifically, we analyzed the depth profiles of the dipole orientation of water molecules, which revealed that the central water layer exhibits a bulk-like structure (see Supplementary Discussion S10). These findings further reinforce our conclusion that interface effects, rather than nanoconfinement effects, dominate the water structure in the nanoconfined 3 L water sample.

## Nanoconfinement effects on water structure

The question arises at what level of confinement the nanoconfinement effects start to appear and modify the water structure so the structure of the water under nanoconfinement deviates from the cooperative interfacial effects. To address this question, we measured the Im$\chi^{(2)}_{yyz}$ spectra on the nanoconfined water sample while tuning the thickness of the nanoconfined water, achieved by changing the RH in the sample cell (see Supplementary Methods)[55,56]. The data are shown in Fig. 3a. As expected, the Im$\chi^{(2)}_{yyz,\,confined}$ spectrum remains unaffected when increasing the water thickness above 8 Å (>3 L water), but it changes substantially when reducing the thickness from ~8 Å to ~5 Å (around two-layer (2 L) water, see AFM data in Supplementary Discussion S5). The negative 3200 cm$^{-1}$ peaks and the positive 3460 cm$^{-1}$ peak decrease dramatically, and the 3670 cm$^{-1}$ peak nearly disappears. Such spectral change is more significant with further decreasing the water thickness down to ~3 Å (around one-layer (1 L) water). Clearly, Eq. (1) is no longer valid for the nanoconfinement below ~8 Å (3 L water). Lowering the RH decreases the thickness of the nanoconfined water and at the same time, increases its salt concentration. We note both the $\chi^{(2)}_{yyz,\,W/G}$ and $\chi^{(2)}_{yyz,\,CaF_2/W}$ are insensitive to ion concentration at high ion concentration (≥2 M), the observed abrupt change of the Im$\chi^{(2)}_{yyz}$ spectra upon decreasing the RH cannot be simply explained by the

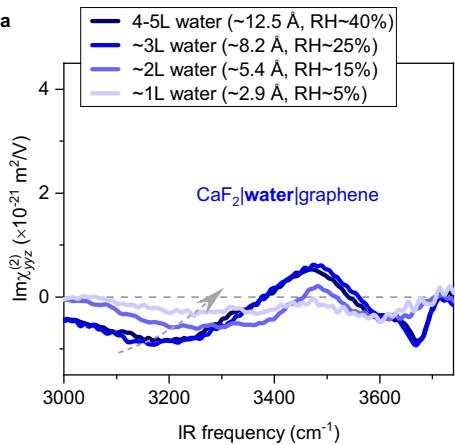
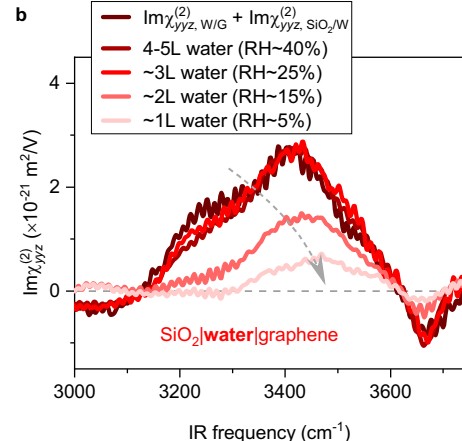
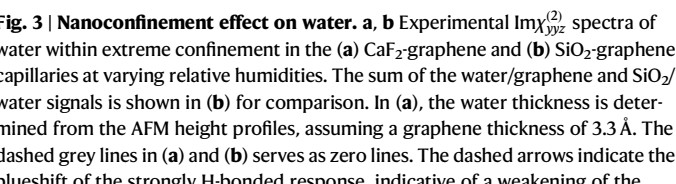

**Fig. 3 | Nanoconfinement effect on water. a, b** Experimental $Im\chi_{yyz}^{(2)}$ spectra of water within extreme confinement in the (**a**) CaF$_2$-graphene and (**b**) SiO$_2$-graphene capillaries at varying relative humidities. The sum of the water/graphene and SiO$_2$/water signals is shown in (**b**) for comparison. In (**a**), the water thickness is determined from the AFM height profiles, assuming a graphene thickness of 3.3 Å. The dashed grey lines in (**a**) and (**b**) serves as zero lines. The dashed arrows indicate the blueshift of the strongly H-bonded response, indicative of a weakening of the

H-bonds. Note that the net orientation of water remains reversed between the two systems, even for the strongest confinement−with negative and positive $Im\chi_{yyz}^{(2)}$, for CaF$_2$-graphene and SiO$_2$-graphene capillaries, respectively. Data in (**a**) were obtained from two independent samples per RH condition. Data in (**b**) were obtained from a single sample measured under different RH conditions. Source data are provided as a Source Data file.

increase of ion concentration in confinement (See Supplementary Discussion S1 and S11 for more details). Rather, these substantial spectral changes indicate that the nanoconfinement effects start to appear and gradually dominate. The decrease of the two H-bonded O-H peaks and the disappearance of the 3670 cm$^{-1}$ dangling O-H peak imply that the <8 Å confinement forces the nanoconfined water to lie flat with O-H groups parallel to the graphene and the CaF$_2$ substrate, consistent with recent theoretical predictions[4,41]. The parallel-orientated O-H groups yield zero SFG signals so the peak intensities decrease. The overall shift of the spectral intensity to higher frequencies (bule-shift) denotes that the H-bond network is weakened substantially due to the enhanced confinement. The HD-SFG measurements on different thicknesses of the nanoconfined water demonstrate that the nanoconfinement effects on the water appear and dominate when the confinement reaches the angstrom scale (<-8 Å, 3 L water, see Supplementary Discussion S12−15 for more simulation data).

The substantially weakened H-bonding network in nanoconfined water as the degree of confinement increases within the CaF$_2$-graphene capillary is notable (Fig. 3a). A similar trend is observed in the SiO$_2$-graphene capillary (Fig. 3b). Our SFG spectra further reveal the reorientation of water molecules under extreme confinement, specifically in conditions with confinement below two water layers. Remarkably, as seen in Fig. 3, we observe significant down-alignment of water molecules within the CaF$_2$-graphene capillary and pronounced up-alignment within the SiO$_2$-graphene capillary. Previous theoretical predictions for hydrophobic confinement, such as between two hydrophobic walls (e.g., hBN and graphene), suggest a vanishing up/down alignment of confined water molecules under similar confinement conditions[41]. Our experimental results highlight that the chemical nature and polarity of the confining walls retain a strong influence on the net orientation of water molecules, even under extreme confinement. Notably, the molecular orientation is not just a structural feature−it plays a pivotal role in modulating water's dielectric properties, as the alignment of water molecules directly affects their collective dipole response[8,64,77,78]. These findings underscore the intricate interplay between molecular orientation, surface interactions, and dielectric properties, shedding light on understanding the anomalous dielectric behavior of water under extreme confinement[8,28].

## Discussion

Our work addresses the fundamental question of the nanoconfinement effect on water, and sheds new light on weak (≥-8 Å, 3 L water) and strong (<-8 Å, 3 L water) confinement. For weakly nanoconfined two-dimensional water, the molecular structure (orientation and H-bond network) differs significantly from bulk water, consistent with previous experimental observations that water properties deviate from bulk already at modest confinement (-2−10 nm)[8,22,24,26−36]. These previous studies employed ensemble measurements, such as conductance[24], dielectric[8], and force measurements[32,35], while these methods provided valuable information on nanoconfined water properties, they offered relatively limited insights at the molecular level. Understanding these molecular details is essential for fully comprehending the properties of nanoconfined water. Our results with resolved molecular details, including its orientation and hydrogen-bond network, suggest that the non-bulk-like response of weakly nanoconfined water observed in these previous studies may primarily arise from cooperative interfacial effects rather than nanoconfinement effects. Nanoconfinement effects on water structure only become significant when confinement is reduced to the angstrom scale (<-8 Å, 3 L water).

The role of interface and confinement effects on the structure of water within two-dimensional confinement has been extensively studied through various theoretical and computational simulation approaches[40−46], making it a topic of ongoing interest. These studies have analyzed key properties such as hydrogen-bonding networks and dipole orientation. Distinct confinement- and interface-dependent regimes have been proposed, often characterized by a critical water thickness of 2−4 layers. Building upon these well-established theoretical frameworks, our experimental study serves as a critical validation and extension, offering direct observations of these regimes under realistic conditions and bridging the gap between simulation and experimental results. Note that our conclusions were derived based on concentrated electrolytes (on the order of molar concentrations). Nanoconfined aqueous systems with molar-range ion concentrations are common in both natural and technological contexts, where nanoscale confinement occurs at charged interfaces such as minerals (e.g., CaF$_2$[61,79], SiO$_2$[24], and mica[55]), two-dimensional materials (e.g., graphene oxide[80], hBN[28,81]), polymers[82], and biological systems (e.g., aquaporins[83], ion channels[84]). In these systems, local charge neutrality

requires high counterion concentrations in the molar range, governed by the Donnan equilibrium[70]. Therefore, our focus on nanoconfined water at high ionic strength reflects a broadly relevant and realistic scenario. For dilute electrolytes at charged interfaces, long-range electrostatic interactions can extend over tens or even hundreds of nanometers (known as Debye screening length)[85]—distances significantly larger than the proposed ~1 nm threshold for the onset of nanoconfinement effects on water structure. In this case, effects from long-range electrostatic interactions, such as electrical double-layer overlap, become relevant, as previously discussed[86,87], but are beyond the scope of this study.

The properties of water are governed by its hydrogen bond network and the ongoing structural evolution of this network on the picosecond time scale. While we can confidently assert the dominant role of interface effects in shaping the structure of nanoconfined 3 L water—particularly in terms of its molecular orientation and H-bond network—we cannot rule out the possibility that nanoconfinement effects may already influence its dynamics. For example, atomistic simulations have shown that the central water layer in a 3 L system may exhibit slower reorientation or even dynamic arrest compared to bulk water[88,89]. This suggests that nanoconfinement may impose a higher threshold for water dynamics under such conditions. The interplay between interface effects and confinement effects on the dynamics of nanoconfined water has been extensively studied[16–22]. Milestone experimental work by Fayer et al. [16,17,20–22] has demonstrated that water molecules become dynamically arrested when confined within reverse micelles with diameters smaller than 2.5 nm, highlighting the critical role of confinement. Beyond this threshold, interface effects take precedence in governing the dynamics of nanoconfined water. More recently, experiments have confirmed a consistent threshold of ~2 nm for dynamics of water confined in two-dimensional systems[29]. Building on these studies of nanoconfined water dynamics, our investigation of its molecular structure provides a unified framework for understanding how interface and confinement effects collectively influence the properties of nanoconfined water.

Our insights into the interfacial effects and nanoconfinement effects on water structure, spanning nanoscale to angstrom-scale confinement, have several implications. First, the awareness that interfacial effects govern the structure of nanoconfined water beyond ~8 Å (3 L water) confinement highlights the potential to manipulate its structure and properties through interface engineering. This is particularly evident from the markedly different net arrangements of nanoconfined 3 L water within CaF$_2$-graphene and SiO$_2$-graphene capillaries. Those findings suggest that engineering the interfacial properties of confining materials may be sufficient for technologies relying on nanoscale confinement when confinement does not enter the sub-nm regime, opening avenues for interdisciplinary innovations in energy storage, water treatment, and nanotechnology. Typical examples include the design of porous electrodes with enhanced effective surface area and enhanced specific ion adsorption capacity to improve battery electrochemical capacitance[7,90,91], as well as the development of porous membranes with customized hydrophilicity or hydrophobicity to boost the efficiency of water desalination and purification technologies[92,93]. Furthermore, materials with tailored surface charging and discharging properties could enable the development of advanced nanofluidic devices[23,94], such as artificial neurons with tunable memory and synapse-like dynamics, expanding the potential of neuromorphic computing in aqueous systems. More generally speaking, the awareness of the crucial role of interfacial effects on nanoconfined water (≥8 Å) connects the realms of interfacial chemistry and chemistry in confinement. This connection opens exciting avenues for the design and precise manipulation of nanoscale electrochemical systems[95,96], where the interplay between nanoconfined water and surface properties could further redefine functionalities in energy storage, catalysis, and nanofluidics.

Looking ahead, the observation that nanoconfinement effects on water only emerge under angstrom-scale confinement (<~8 Å, or approximately two layers of water) highlights the necessity of creating such extreme confinement to induce true confinement-induced structural changes in water. Intriguingly, the true nanoconfinement effects observed in angstrom-scale confinement reveal weakened hydrogen bonding strength and altered molecular orientation, strongly influenced by the chemical nature and polarity of the confining walls. This highlights the potential of engineering the chemical properties of confining walls as a rational design principle to enhance reactivity and tailor the selectivity of chemical transformations even under extreme geometric confinement[25]. Notably, nanoconfined water has emerged as a valuable system for investigating water's structural and dynamic behavior in the deeply supercooled regime (e.g., 150–230 K at ambient pressure), where bulk water rapidly crystallizes into ice. Within this context, the distinct interfacial effects and nanoconfinement effects observed across different confinement scales provide critical insights into rationalizing the behavior of supercooled water in nanoconfined environments[97]. Finally, these findings, combined with the advanced experimental capabilities established in this work, open exciting opportunities for future studies exploring the molecular details of water structure and properties, as well as relevant aqueous chemistry and solvation science, under conditions of extreme confinement.

## Methods

### Sample Preparation

The nanoconfined water samples were prepared by using the wet transfer technique[2,98] to enclose water between a graphene sheet and a CaF$_2$ substrate (Fig. S1, see Supplementary Methods in the Supplementary Information for more details). In brief, we used a commercial CVD-grown large-area monolayer graphene sheet on copper foil (Grolltex Inc). The graphene sheet was exposed to LiCl solution and was then transferred onto the CaF$_2$ substrate. A thin film of LiCl solution was trapped between the graphene and the substrate due to the hydrophilic nature of the CaF$_2$ substrate. We prepared samples using KCl, and all reported results were similar (see Supplementary Discussion S1 for more details). Additionally, we fabricated capillary devices with graphene and SiO$_2$ and found the same conclusion (see Supplementary Discussion S1). Upon drying for ~12 h in air with an RH of ~25%, nanoconfined water samples were obtained as a result of capillary condensation[55] (see Supplementary Discussion S5 for more details). The preparation of the suspended graphene on the water surface was similar to refs. 65,99. and was detailed in our recent work[69]. Details of the suspended graphene preparation are provided in Section 4 of the Supplementary Methods of the Supplementary Information.

### HD-SFG measurement

HD-SFG measurements were performed on an HD-SFG setup in a non-collinear beam geometry with a Ti: Sapphire regenerative amplifier laser system, operating at 800 nm central wavelength with ~40 fs pulse width, 5 mJ pulse energy, and a 1 kHz repetition rate. A detailed description can be found in refs. 48,61,100. In brief, the narrowband visible beam (~10 cm$^{-1}$ FWHM) was generated from a part of the laser output using a grating-based pulse shaper with a cylindrical lens. To generate the broadband infrared (IR) pulse (bandwidth ~530 cm$^{-1}$ FWHM), the remaining laser output was directed to an optical parametric amplifier (TOPAS-C, Light Conversion) equipped with a silver gallium disulfide (AgGaS$_2$) crystal. To generate the local oscillator (LO) signal for heterodyne detection, the IR and visible beams were initially focused onto a 200 nm-thick ZnO film coated on a 1 mm-thick CaF$_2$ substrate, using a procedure consistent with previously established approaches[100]. The generated LO, along with the IR and visible beams, were then directed and refocused using two pairs of off-axis parabolic mirrors to achieve spatial and temporal overlap at the nanoconfined

water sample. The incident angles for the IR, visible, and LO beams were (in $CaF_2$) 33°, 39°, and 37.6°, respectively. The measurements were performed at the *ssp* polarization combination, where *ssp* denotes *s*-polarized SFG, *s*-polarized visible, and *p*-polarized IR beams.

HD-SFG spectra were measured in a dried air atmosphere to avoid spectral distortion due to water vapor. The sample cell was purged with $N_2$ of different RHs during the measurement to tune the thickness of the nanoconfined water (see Supplementary Methods). We checked the sample height with a displacement sensor (CL-3000, Keyence with a resolution of -0.1 μm). More details about the sample preparation, sample cell, Raman measurement, HD-SFG measurement, and data analysis can be found in Sections 1–9 in the Supplementary Method of the Supplementary Information.

### AFM measurement
The surface morphology of the graphene samples was measured using an atomic force microscope (AFM, Bruker, JPK) working in the non-contact mode. We used a silicon cantilever (OLYMPUS OMCL-AC 160, $f = 243$ kHz, $k = 25$ N m$^{-1}$) for the measurement. We control the RH inside the AFM chamber by purging the chamber with $N_2$ of different RHs.

### SFG spectral simulation
We performed the MD simulation at the ab initio level of theory with the help of machine learning, and we computed the SFG spectra based on the obtained MD trajectories. We performed the MD simulations at the water/graphene interface and the $CaF_2$/aqueous LiCl solution interface as well as the confined water systems using varying thicknesses of the confined water. The AIMD simulations were carried out by using the CP2K code[101]. From the AIMD data, we created the force field through machine learning technique using the DeepMD-kit package[102]. The cell size of the graphene/water interface was composed of $\vec{a}$, $\vec{b}$, and $\vec{c}$, while the cell sizes of the $CaF_2$/water interface and nanoconfined systems were $2\vec{a}$, $2\vec{b}$, and $\vec{c}$, where $\vec{a} = (14.76$ Å, 0 Å, 0 Å$)$, $\vec{b} = (7.38$ Å, 12.78 Å, 0 Å$)$, and $\vec{c} = (0$ Å, 0 Å, 50 Å$)$. The MLFF-MD simulations were performed in the NVT ensemble with the target temperature of 300 K by using the LAMMPS code[103]. Using the coordinates and velocities of water molecules, we computed the SFG spectra within the surface-specific velocity-velocity correlation function formalisms[104]. Details of the simulation can be found in Sections 10-12 in the Supplementary Method of the Supplementary Information.

## Data availability
The data corresponding to the figures, input files to reproduce the AIMD, and the models to reproduce DPMD simulations are available in ref. 105. The training set of the MLPs is provided in ref. 106. Part of the data, methods, and analysis presented in this article were previously included in the doctoral thesis of Dr. Y. Wang[107], submitted to the University of Amsterdam in 2024. This article also contains new data and analyses not reported in the thesis. Source data are provided with this paper.

## Code availability
All the codes used in this work are freely available and can be found in the corresponding references provided in Methods.

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

## Acknowledgements

We are grateful for the financial support from the MaxWater Initiative of the Max Planck Society. Funded by the European Union (ERC, n-AQUA, 101071937). Views and opinions expressed are, however, those of the author(s) only and do not necessarily reflect those of the European Union or the European Research Council Executive Agency. Neither the European Union nor the granting authority can be held responsible for them. We thank Johannes Hunger, Nikita Kavokine, and Maksim Grechko for providing insightful comments and suggestions on this work. We thank Ruediger Berger for providing support on AFM measurements. We also thank Florian Gericke, Marc-Jan van Zadel, and the technical workshop at the Max Planck Institute for Polymer Research for excellent technical support. F.T. is supported by National Key R&D Program of China (Grant No. 2024YFA1210804) and a startup fund at Xiamen University. Part of this work used the computational resources in the IKKEM intelligent computing center and the ISSP at the University of Tokyo.

## Author contributions

Y.K.W., Y.N., and M.B. designed the study. Y.K.W. prepared the samples and performed the AFM and Raman microscope measurements. Y.K.W., X.Q.Y., K.Y.C., and C.C.Y. performed HD-SFG measurements. T.O. and Y.N. conducted the AIMD simulations. F.T. conducted the MLFF-MD simulations and computed the SFG spectra. The data analysis was done by Y.K.W. (experiment) and F.T. (simulation). Y.K.W. and Y.F.C. contributed to discussions on sample preparation. Y.K.W., F.T., Y.N., and M.B. wrote the manuscript. All authors contributed to interpreting the results and refining the manuscript.

## Funding

## Competing interests

The authors declare no competing interests.
