## [Transparent Peer Review file · Nature Communications]

Interfaces Govern the Structure of Angstrom-Scale Confined Water Solutions

Corresponding Author: Professor Mischa Bonn

Version 0:

Reviewer comments:

Reviewer #1

(Remarks to the Author)

After evaluating this manuscript now for the third time, I continue to stand to my previous recommendation to accept it for publication, now in Nature Communications.

The authors have again constructively addressed the feedback from the previous round.

In particular, I want to comment on the argument about the validity of AIMD for describing these questions. As an expert on DFT, AIMD, and MLMD, I can only agree with the assessment of the authors that these methods are state of the art and perfectly suited as counterpart for the experimental measurements.

AIMD is perfectly suited to describe the "complex Coulomb many-body physic" of these systems, given that it solves the electronic Schroedinger equation on-the-fly during the dynamics.

Reply to Reviewer #1:

Reviewer #1 (Remarks to the Author):

After evaluating this manuscript now for the third time, I continue to stand to my previous recommendation to accept it for publication, now in Nature Communications.

The authors have again constructively addressed the feedback from the previous round.

In particular, I want to comment on the argument about the validity of AIMD for describing these questions. As an expert on DFT, AIMD, and MLMD, I can only agree with the assessment of the authors that these methods are state of the art and perfectly suited as counterpart for the experimental measurements.

AIMD is perfectly suited to describe the "complex Coulomb many-body physic" of these systems, given that it solves the electronic Schroedinger equation on-the-fly during the dynamics.

Reply: We sincerely thank you for your thorough and constructive evaluation of our manuscript throughout the review process. We are pleased that our responses have addressed your concerns, particularly regarding the validity of AIMD in capturing the complex Coulomb many-body physics. We also greatly appreciate your positive recommendation to accept our work for publication in *Nature Communications*.